# Behavioral and Autonomic Responses in Treating Children with High-Functioning Autism Spectrum Disorder: Clinical and Phenomenological Insights from Two Case Reports

**DOI:** 10.3390/brainsci10060382

**Published:** 2020-06-17

**Authors:** Lucia Billeci, Ettore Caterino, Alessandro Tonacci, Maria Luisa Gava

**Affiliations:** 1Institute of Clinical Physiology, National Research Council of Italy, 56124 Pisa, Italy; lucia.billeci@ifc.cnr.it; 2Azienda USL Sudest Toscana, Centro Autismo UFSMIA di Grosseto, Ospedale di Castel del Piano, 58033 Grosseto, Italy; ettore.caterino@uslsudest.toscana.it; 3Associazione Nazionale Famiglie di Persone con Disabilità Intellettiva e/o Relazionale (ANFFAS), 18100 Imperia, Italy; marialuisa.gava@gmail.com

**Keywords:** high-functioning autism, language, experience, communication, autonomic nervous system, wearable technologies

## Abstract

In this study, we aimed to evaluate the process applied in subjects with Autism Spectrum Disorder (ASD) to elaborate and communicate their experiences of daily life activities, as well as to assess the autonomic nervous system response that subtend such a process. This procedure was evaluated for the first time in two eight-year-old girls with high-functioning ASDs. The subjects performed six months of training, based on the cognitive–motivational–individualized (c.m.i.^®^) approach, which mainly consisted in building domestic procedures and re-elaborating acquired experiences through drawing or the use of icons made by the children. Together with behavioral observations, the response of the autonomic nervous system during such re-elaboration was recorded. A change in communicative and interactive competences was observed, moving from a condition of spontaneity to one in which the girls were engaged in relating their experiences to a parent. Autonomic response highlighted how, in communicating their own experiences, they achieved a state of cognitive activation, which enabled a greater communicative and emotional connection with the interlocutor. This is a proof-of-concept study on the application of the c.m.i.^®^, which needs to be extensively validated in the clinical setting.

## 1. Introduction

In the field of Autism Spectrum Disorders (ASD), several studies showed the importance of structuring a space, a time and a specific visual support as part of the related clinical intervention [1]. Many behavioral approaches provide a visual target to a child with ASD in order to implement the acquisition of a skill (i.e., washing the hands), assuming that the child would be more sensitized and would have a greater level of attention and orientation if induced to the object or action by an iconic structure.

However, according to our previous research study [2], our idea is that a child with ASD, somewhat similar to a typical developing child, should learn a procedure in a natural context (i.e., in their own environment) and then, eventually, use icons or visual references to recall the acquired skills. In this way, the child will be able to understand the real meaning of the visual reference used with respect to what the children have acquired during their own experience. In this way, the icons will only represent the starting point for recalling the actions the subject has previously learned (i.e., washing, dressing, cooking, etc.). Notably, this process requires the subject to be able to manipulate the experience acquired with a higher level of awareness and participation and then to verbalize and recall their own experiences. To recognize their experience, they should mentally manipulate the acquired contents (i.e., who, what, where and when) and the connections made among the motor actions performed. This is often difficult in children with high-functioning ASD due to the presence of deficits in executive functions [3,4], in social perception [5] and in narrative and pragmatic skills.

In this framework, the cognitive–motivational–individualized (c.m.i.^®^) approach is aimed at supporting an orientation and awareness process of personal reality and knowledge in subjects with verbal and/or cognitive and/or relational disabilities, so that they can organize and express their thoughts in a more understandable and orderly way. This approach, initially set up to facilitate work with augmentative and alternative communication (AAC) [6], represents a learning process since, through a specific reconstruction of daily life activities and work, the subject is helped to acquire these experiences in a more conscious and autonomous way. Specifically, to facilitate this process, a figurative reworking is proposed in order to make the experience concrete, visible and manually usable. From this perspective, the subject must retrace the practical process of the experience both on a symbolic plane and on a motor level, manipulating the components (i.e., what, where, who) and their connections or semantic constraints (i.e., what/where, who/what) in a praxis narrative form that refers to the experiential action. This would enable an understand of the process within which the subject retains the elements that are significant for them in reality (what he/she knows, what he/she does, what he/she likes, where the experiences take place and with whom they do these experiences) and how performing motor actions (acting on the images, the icons drawn from his/her everyday experience) re-elaborates his/her knowledge. The overall aim is to facilitate the integration between the experiential and phenomenal meaning performed with the body and language (lexical semantics) to avoid—as often happens in children with ASD—that, during the narration, the subject is only using the lexical meaning without the emotional content, which defines intentionality in the communication, the participation and, therefore, the intersubjective sense of language [7,8,9].

Together with behavioral observation, the assessment of physiological parameters during the task of reproducing the acquired experience in ASD subjects could help the clinician to understand the way the subjects are processing these experiences. In particular, several studies suggest that Autonomic Nervous System (ANS) activity is linked to social functioning in individuals with ASD [10,11,12,13,14]. Both the sympathetic (SNS) and parasympathetic (PNS) branches of the ANS contribute to social functioning and communication [15,16], with SNS activation reflecting a threat-oriented response, and PNS dominance easing adaptive social engagement [17]. More specifically, the PNS plays an important role in social functioning by innervating several organs in the face and in the neck and affecting various functions, including cardiac activity, which are related to social behavior and communication [18]. Thus, in a social setting, an increase in the PNS activity is related to an adaptive social behavior [13]. The SNS also plays an important role in social functioning, increasing heart rate (HR), sweating and alert state, mainly through acting on the so-called “fight or flight” mechanism [19,20].

Wearable technologies can be particularly relevant in the assessment of ANS in ASD due to their low obtrusiveness. In particular, the recording of the electrocardiogram (ECG) signal allows us to evaluate the activation of both the PNS and SNS by studying heart rate variability (HRV) [10]. On the other hand, the assessment of the galvanic skin response (GSR) is mainly related to the SNS, with a reduction in the signal associated with decreased SNS influence [21], whereas an increase in skin conductance during specific social tasks subtends higher SNS activation [22].

In this study, we aim to highlight how, in subjects with high-functioning ASD, it is a priority to enhance the mechanisms of self-awareness related to their experiences and knowledge rather than immediately using linguistic codes, icons or scripts or giving heuristic value to a phenomenological approach. More specifically, the objectives of the present study include: (i) to evaluate the behavior and communication of two subjects with ASD during the elaboration of an experiential content (procedural actions of shared daily life with the parent); (ii) to assess the ANS responses that underlie the subjects’ behavior during such elaboration.

## 2. Materials and Methods

### 2.1. Participants

The subjects were two eight-year-old girls, with diagnoses of ASD, attending the “Autism service of the public health of the city of Grosseto”. The protocol was approved by the Institutional Ethical Clearance board of the National Research Council (0087922/2019).

#### 2.1.1. Subject 1—J

At the age of 3, J. received a clinical diagnosis of Autism Spectrum Disorder (ASD) confirmed with the Autism Diagnostic Observation Schedule-Generic (ADOS-G) [23]. Her non-verbal cognitive level, measured through Leiter-R [24] and Color Progressive Matrices [25] was average. In her clinical history, motor and language delays were reported. At the age of 3, J. showed a severe psychomotor and language regression. In this period, J. also partially lost sphincter control, showing sialorrhoea and feeding disturbance. She was severely apathetic. Before enrollment in this study, she started a psychoeducational intervention including speech and psychomotor therapies. At the age of the enrollment (6 y), J. presented a normostructured language with flat prosody and a socio-pragmatic disorder. Regarding her motor skills, she presented hypotonia and coarse-motor impairment, while she had adequate fine motor praxes and graph-pictorial skills.

#### 2.1.2. Subject 2—B

At the age of 5, B. received a clinical diagnosis of Autism Spectrum Disorder (ASD) confirmed with ADOS-G. Her cognitive level, measured through Wechsler Preschool and Primary Scale of Intelligence (WPPSI-III) [26], was average (Full Scale Intelligence Quotient, FSIQ = 107). In her clinical history, language delay, marked issues in emotional dysregulation and Attention Deficit Hyperactivity Disorder (ADHD) were reported. Before enrollment in the study, she started a psychoeducational intervention. At the age of enrollment (6 y), B. was an hyperverbal child with high functioning ASD in comorbidity with ADHD and emotional dysregulation (i.e., meltdown). She used reinforcement strategies for performing school—and daily life—activities (i.e., planned and structured environments and activities, visual aids).

### 2.2. The c.m.i.^®^ Approach

The c.m.i.^®^ approach adopts a naturalistic and developmental approach (NBDI) [27]. It targets social, interactive and communication impairments in autism. The rationale is that children with autism would respond with enhanced communicative and social development to a style of parent communication adapted to their impairments [28]. The intervention is aimed to enhance the link between neuropsychological and relational skills. The rationale of c.m.i.^®^ is the application of embodied cognition in the intervention for ASD. Inspired by Johnson-Glenberg and Megowan-Romanowicz [29], the three degrees of embodiment are considered: (a) sensorimotor experience; (b) link between body language and feelings; (c) emotional immersion experienced by the user.

The c.m.i.^®^ methodology is aimed at supporting an orientation process and awareness of one’s reality and knowledge in subjects with verbal, cognitive and relational disabilities so that they can organize and express their thoughts in a more comprehensible and orderly way. This experience is made explicit initially using the body—doing and perceiving also represent a form of non-verbal communication which subsequently and progressively evolves into language. The intervention allows a progressive maturation of orientation abilities in reality that affects the following dimensions: (i) spatial: where are things, people, objects; (ii) temporal: when events occur; (iii) instrumental: objects in the environment, their nature, their functions; (iv) relational: people around the subject.

### 2.3. Training According to the Cognitive Motivational Intervention (c.m.i.^®^)

The training of the two subjects according to the c.m.i.^®^ approach was performed at home and involved the parents. It consisted of the following phases:Interview with the parents: During the interview, the parents were informed about the aims and modalities of the training approach and their involvement in the process. In addition, informed consent was obtained from the parents of the two girls enrolled, after receiving an exhaustive explanation of the study. Such explanation included: (a) the description of the theory of the cognitive motivational intervention; (b) the rationale of physiological signal acquisition and the procedures for data collection; (c) detailed description of home procedures; (d) the description of the re-elaboration of acquired experiences through drawing; (e) the modalities of using icons by the children.Observation of the two girls: This aimed at identifying the individual characteristics of the two girls. J. loves drawing and adds plenty of particulars in her drawings, although images look quite static. Conversely, B. is constantly in motion and uses continuous, redundant, sophisticated and dyspragmatic language. She does not like drawing, but rather prefers moving and talking. She has difficulties in maintaining task concentration. In order to manage individual differences among the subjects, meetings with parents were organized in order to explain to them the individual phenotypic profile of their child and how to manage the emotional experience of the child as well as their own. Specifically, therapists contacted the parents four times a month after the initial meeting.Training to realize the procedures: The parents were instructed how to guide the two girls in the acquisition of specific procedures, which consisted in cooking the girls’ favorite foods (spaghetti for J. and biscuits for B.). Then, the girls were helped in acquiring information about the environments in which they were supposed to cook (which objects to use and where they are located). Once a week, the two girls were involved in making up their favorite food.

### 2.4. Evaluation of the Acquired Procedure

The evaluation of the acquired procedure was performed in a laboratory setting and was aimed at recalling the procedures acquired at home to allow the transition from praxis to a symbolic action when the girls reached a certain degree of autonomy. The session was guided by the experimenter who developed the c.m.i.^®^ approach (M.L.G.).

The first phase of the evaluation (after six months) consisted in performing a graphical trace, which represented a symbolic reconstruction of the kitchen and objects used at home. Each girl was seated next to the experimenter at a table on which some papers and markers were positioned. First, the girl was guided in recovering the experience learned at home on the verbal level and subsequently in the drawing of the objects and the ingredients she used at home for cooking the selected food.

The second phase of the evaluation (after 10 months) consisted in the action on the represented, which dealt with the description of the experience acquired at home through the use of mobile icons to reproduce the motor action performed in the real scenario.

### 2.5. Physiological Signal Acquisition

During the evaluation procedure, the two subjects were equipped with two minimally obtrusive, wireless, wearable sensors for the monitoring of the electrocardiogram (ECG) and galvanic skin response (GSR). The ECG signal was acquired through the Shimmer ECG sensor (Shimmer Sensing, Dublin, Ireland) with a sampling frequency of 500 Hz, whereas the GSR signal was captured by the Shimmer3 GSR+ sensor (Shimmer Sensing, Dublin, Ireland) with a sampling frequency of 51.2 Hz. The Shimmer ECG acquired the relevant signal attached to a fitness-like chest strap manufactured by Polar Electro Oy (Kempele, Finland), whereas the Shimmer3 GSR+ captured the galvanic skin response by adhering to two nearby fingers of the subject’s non-dominant hand through comfortable rings. The signals were acquired while the subject was seated on a chair at a table during the following phases:Baseline (3′): Basal measurement. Here, the subject was asked to stay as still as possible.Task (15′): Signal acquisition during the proposed tasks.Recovery (3′): Post-task basal measurement. The subject was required to stay as still as possible, similar to baseline.

### 2.6. Physiological Signal Analysis

The acquired physiological signals were processed using Matlab (Mathworks, Natick, MA, USA). In particular, the ECG signal was analyzed using home-made scripts for the calculation of the tachogram (RR series; i.e., the time elapsed between two successive R-waves) according to the Pan–Tompkins algorithm [30] and to extract both time and frequency domain features characterizing the ANS for each of the different phases.

The features extracted included:Heart rate (HR): the number of contractions of the heart occurring per time unit, expressed in bpm;Standard deviation of normal to normal RR intervals (SDNN): measure of heart rate variability (HRV), expressed in ms;Normalized component of the power spectral density of the ECG signal at low frequency (0.04–0.15 Hz) (nLF), which is related both to the sympathetic and parasympathetic response;Normalized component of the power spectral density of the ECG spectrum at high frequency (0.15–0.4 Hz) (nHF), which is mainly related to the parasympathetic response;Low versus high frequency components of the power spectral density of the ECG spectrum (LF/HF Ratio), which expresses the balance between the sympathetic and parasympathetic nervous system branches.

GSR signal was analyzed using Ledalab V3.4.9 (General Public License (GNU)), a Matlab-based tool enabling the user to extract typical features of the GSR signal, notably its tonic and phasic components [31]. Specifically—albeit both of the abovementioned components were extracted—for this work only the tonic phase was analyzed, since the rationale of the study aimed at comparing the signal in the different phases rather than evaluating the response to a given single stimulation.

Additional details about the analysis procedure can be found in previous studies [32,33].

## 3. Results

To assess the effects of treatment for every child and, keeping in line with the case series design, data were examined on an individual participant basis. Individual changes in social communication, adaptive behavior and neural response to social stimuli are discussed below.

### 3.1. Behavioral Observations

After a six month period (follow-up) of behavioral observations of the two subjects included in the present work, a critical evaluation was conducted concerning the experience carried out at home.

J. loved drawing. Compared to her previous drawings that were very static, she added an increasing number of dynamic characteristics in her storytelling and her drawings (i.e., climbing stairs, drying clothes). She was very precise and participated in the reconstruction of the objects, using language, eye contact and checking whether the therapist or the adult was able to listen to her and whether her work was being appreciated, as shown in Figure 1a.

B. reported difficulties in terms of the graphical realization of the tasks demanded. She preferred to use her voice to describe the steps of the recipe learned at home, while the experimenter realized the drawing, as shown in Figure 1b. She displayed quite rich and cogent language, albeit not always adherent to reality. The deposition of her mother was particularly interesting, as she reported significantly less automated language and an optimal self-care when physically engaged in actions.

After 10 months (second follow-up), the re-evaluation consisted in the action of the represented task, aimed at representing actions developed at home within the praxis procedure using mobile icons on a re-constructed background. The adult helped the child in re-constructing the kitchen on a paper sheet, positioning fixed objects (including furniture), whereas mobile ones were manipulated to narrate the procedure of cooked food.

J. verbalized the different phases, being very careful in adding elements related to the procedure, as shown in Figure 2a.

B. often used mobile elements with good motor and language competence. Compared to the graphical trace evaluation, she expressed more self-control, as shown in Figure 2b.

Importantly, both girls’ parents seemed to participate in their daughters’ work. They highlighted with surprise and continuous discovery some characteristics of their daughters and they were more attentive about the movements of their children in the environment and about the way actions were followed (e.g., preparing “green” spaghetti and “white and black” biscuits, etc., spontaneously and with personal semantic links between objects and actions).

### 3.2. Autonomic Nervous System Response—Graphical Trace

With J., the task highlighted the presence of verbalization with and without ocular contact. When the drawing was similar to the experienced reality, the subject increased ocular contact with respect to the therapist, whereas a stereotyped drawing drove her to hypo-regulation and relational disinvestment.

According to behavioral observations, for J. the physiological signals were segmented in three different sub-phases during the execution of the task: verbalization with eye-contact, verbalization without eye-contact, and no response (during which she was concentrated on drawing and did not reply to the experimenter).

Concerning the ECG, an increased HR and a decreased HRV during verbalization with ocular contact was observed, as shown in Figure 3a,c. Conversely, during verbalization without eye contact, as well as during non-response phases and less cognitively-demanding tasks, the HRV was significantly increased, as shown in Figure 3c. Focusing on the SNS/PNS activation, it is worth noting that the SNS activity increased during the task, with somewhat of a SNS withdrawal during the verbalization with eye contact, probably due to a decrease in the stress level of the subject when interacting with the therapist, as shown in Figure 3b,d.

With J., the GSR signal displayed an increased phasic component, related to the response to a given stimulation during the task, whereas the tonic component mainly increased at recovery, indicating that the stress response, activated by the task performed, remained active after the completion of the task.

B. verbalized both during the execution of the graphical trace by the therapist and without the drawing. According to this behavioral observation, for B. the physiological signals were segmented in two different sub-phases during the execution of the task: verbalization with the graphical trace and verbalization without the graphical trace.

Concerning the ECG signal, the HR was increased from baseline to task during both phases, probably suggesting a higher arousal level while performing the tasks, followed by a similar decrease during recovery. Similar to the first subject, the HRV decreased here during the task, particularly while verbalizing and performing graphical trace (i.e., during a higher emotionally-demanding task), as shown in Figure 4a,c.

Focusing on the SNS/PNS activation, the second subject already displayed an over-activation of the SNS at baseline, with a PNS dominance during the task, ending with the return to a SNS prevalence at recovery, with the opposite trend with respect to the first individual analyzed, as shown in Figure 4b,d.

The GSR signal revealed an increased tonic and phasic component at task, with further stress evidence at recovery, possibly due to the environmental constraints necessarily applied to complete the experimental task (i.e., sitting on a given chair, staying in a given room, etc.).

### 3.3. Autonomic Nervous System Response—Action on the Represented

In the second task, the action on the represented task, no specific sub-phases could be defined, therefore the signals were not further segmented.

With J., an increased HR was observed with a contextual HRV decrease, probably suggesting significant cognitive involvement by the subject, as shown in Figure 5. Focusing on the SNS/PNS activation, the PNS component was dominating throughout the trial, with a slight increase in SNS activity at task.

The GSR signal showed an increased tonic component at task, probably suggesting an enhanced emotional involvement, whereas the phasic component appeared to be decreased, possibly because of the emotional involvement taking place for the engagement phase more than for the single tasks administered.

With B., a decreased HR and HRV at task was noticed, probably due to relaxation and enhanced task-demanded concentration, as shown in Figure 6.

The evaluation of SNS/PNS confirmed the tendency towards a SNS decrease at task, lasting up to the recovery phase, confirming the regulatory effect of the demanded task, considering the different basal involvement of the subject, who was more stressed at the beginning of the experimental protocol with respect to the previous individual.

Finally, as for the GSR signal, both the tonic and phasic components are higher as long as the test takes place, probably due to the abovementioned logistic constraints for the test administration for the previous subject tested.

## 4. Discussion

The most important lesson learned during the test administration and the discussion of the results of the present study protocol is the pivotal need for the personalization of the therapy that children with neurodevelopmental disorders should undergo. Indeed, despite the fact that the present study took into account just two subjects, with the same age and somewhat similar clinical characteristics, the personal attitudes, behaviors, and past histories surely played a significant role in their responses, with completely different neurophysiological adaptations to the tasks administered displayed by the two children. Indeed, different autonomic responses were seen between the two individuals, with different trends likely to be due to the personal attitudes of both subjects. With J., the clinical observation and the physiological parameters show how the girl, when left alone, tends towards a condition of hypo-regulation and relational closure. In this phase, she draws in a stereotyped way. When asked to design her own experiences, on the contrary, she expresses good communication skills, eye contact, and a greater adherence to the represented content. In this phase, the physiological parameters show that she is more engaged in the task and she has a good relationship with the examiner with the experiences she has learned in her own reality. With B.—who also has a hyper-regulated neurosensory profile at the baseline—progressive emotional regulation, emotional engagement and, therefore, good homeostasis, was obtained through cognitive reorganization (graphic trace, action on the represented task) and the narration of her experiences. Notably she remains regulated in the recovery phase after performing the drawing and the action on the represented task, as if, through this experience of reorganization, she could reach the regulation of her emotional components that usually tend to be dysregulated, impulsive and destabilizing.

These observations of individual characteristics and autonomic responses of the subjects, that happen quite often in presence of subjects with ASD, even with similar clinical characteristics, pave the way to a strongly personalized approach to the treatment.

To this extent, the proof of concept related to the importance of precision medicine in neurodevelopmental disorders has been already highlighted in past years, with interesting links postulated within the framework of genotype–phenotype relationships [34]. Indeed, treatment personalization might provide some benefit by addressing some of the deficits shown by children, even in the field of neurodevelopmental disorders [35].

The present work suggests that the cognitive–motivational–individualized (c.m.i.^®^) approach, the main focus of our investigation, sets its basis on this concept, completely relying on providing a complete overview of the individual aspects of a child and taking into account both cognitive and motivational dimensions to optimize the treatment outcome. In particular, the therapist administering the protocol should be able to empower the positive mindset of the individual treated. Indeed, ASD, even more than other neurodevelopmental disorders, presents individuals that are highly motivated to engage in their special interests, and are more motivated than non-ASD subjects (particularly typically developing controls) by intrinsic motivational factors, some of which are associated with positive effects [36]. Additionally, cognitive level should be taken into account in ASD, as demonstrated by the positive cognitive gains that behavioral and cognitive therapy has on individuals with ASD during adolescence (see [37] for an example) and even during adulthood [38].

Such positive preliminary impressions, albeit necessitating the confirmation on large-scale studies brought forward by the first results of this investigation, should be carefully considered, even in light of some acknowledged limitations. At first, the enrollment of just two subjects does not allow us to draw conclusions applicable to the general ASD population, or even to its subgroups (i.e., High-Functioning ASD); however, it gives us room for further investigation, applying c.m.i.^®^ to larger cohorts likely to confirm an extreme heterogeneity of the treatment response. Notably, this study represents a proof-of-concept for the application of the c.m.i.^®^ approach, which is based on a theoretical framework, and on the assessment of the physiological response of subjects with ASD during such treatment.

Second, the application, in this protocol, of c.m.i.^®^ to only female subjects could reveal cognitive, behavioral and motivational characteristics that are specifically gender-biased, thus not reflecting the reality possibly observed among males, which represent the vast majority of ASD individuals (see [39] for some estimates).

Third, the fact that the two subjects enrolled belonged to an age group between 6 and 10 years, together with the absence of a control group, made it difficult to ascertain the likely positive effect due to c.m.i.^®^ with respect to a general positive effect due to age. Therefore, future studies should include a control group to effectively discriminate further the contribution of c.m.i.^®^ to the personal cognitive growth of ASD children.

Fourth, the use of wearables, even if recently more frequent in the scientific literature dealing with ASD (see, for example, [40,41]), should take advantage of a wider, better grounded amount of data, which will be possible step-by-step in next few years when such methodologies, still relatively new in the neurophysiological monitoring of ASD, will fully enter scientific society.

Under such premises, still taking into full consideration the abovementioned limitations, and if confirmed on large-scale studies, the proposed approach could flank traditional therapies, possibly improving the patients’ outcomes based on their own specificities, paving the way for precision medicine in ASD and neurodevelopmental disorders. Further studies are needed to extensively apply and validate the proposed approach in a clinical setting.

## 5. Conclusions

The pilot study described here presented a novel approach for improving cognitive features of children and adolescents with ASD, leaning on personal motivations and own interests. Despite presenting some benefits for the patients included in the present study, the beneficial effects of this approach should be confirmed on larger cohorts, also taking advantage of the enrollment of control groups, lacking in our protocol. Eventual, likely positive results would further enable the large-scale application of the c.m.i. approach flanking traditional cognitive and behavioral therapies.

## Figures and Tables

**Figure 1 brainsci-10-00382-f001:**
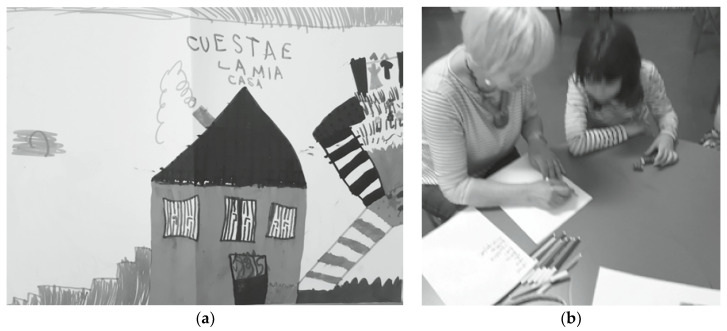
The first evaluation consisting in the graphical trace. (**a**) The drawing by J., (**b**) B. watches the experimenter drawing while she verbalizes.

**Figure 2 brainsci-10-00382-f002:**
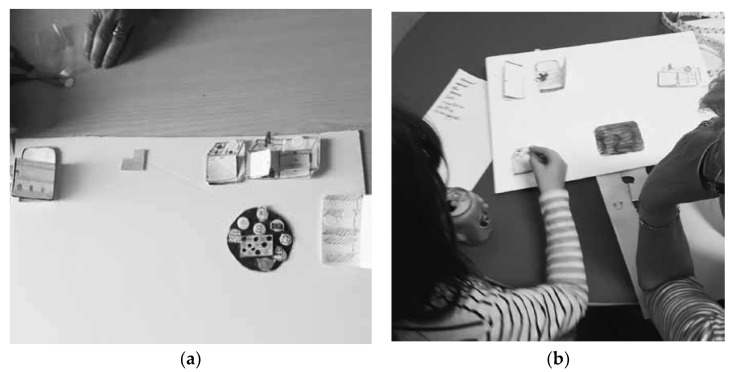
The second evaluation consisting in the action on the represented task. (**a**) The cooking of the biscuits realized with the icons by J., (**b**) B. reconstructing the procedure of cooking spaghetti.

**Figure 3 brainsci-10-00382-f003:**
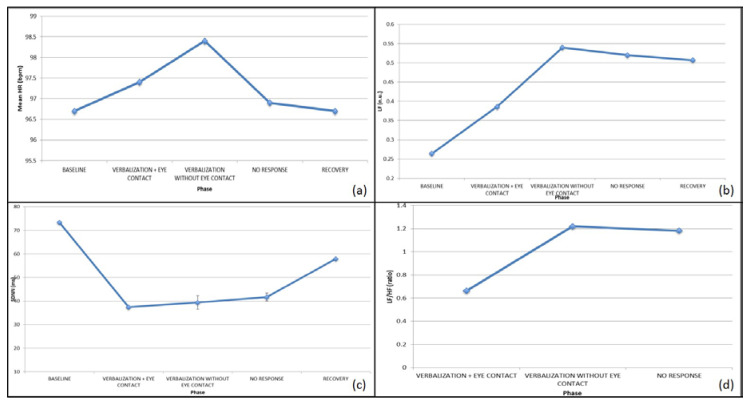
ECG features for J. during the various phases of the graphical trace: (**a**) HR, (**b**) LF, (**c**) SDNN, (**d**) LF/HF.

**Figure 4 brainsci-10-00382-f004:**
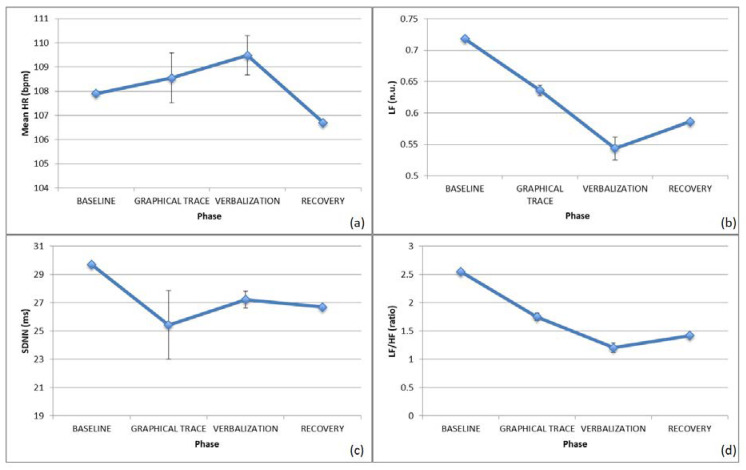
ECG features for B. during the various phases of the graphical trace: (**a**) HR, (**b**) LF, (**c**) SDNN, (**d**) LF/HF.

**Figure 5 brainsci-10-00382-f005:**
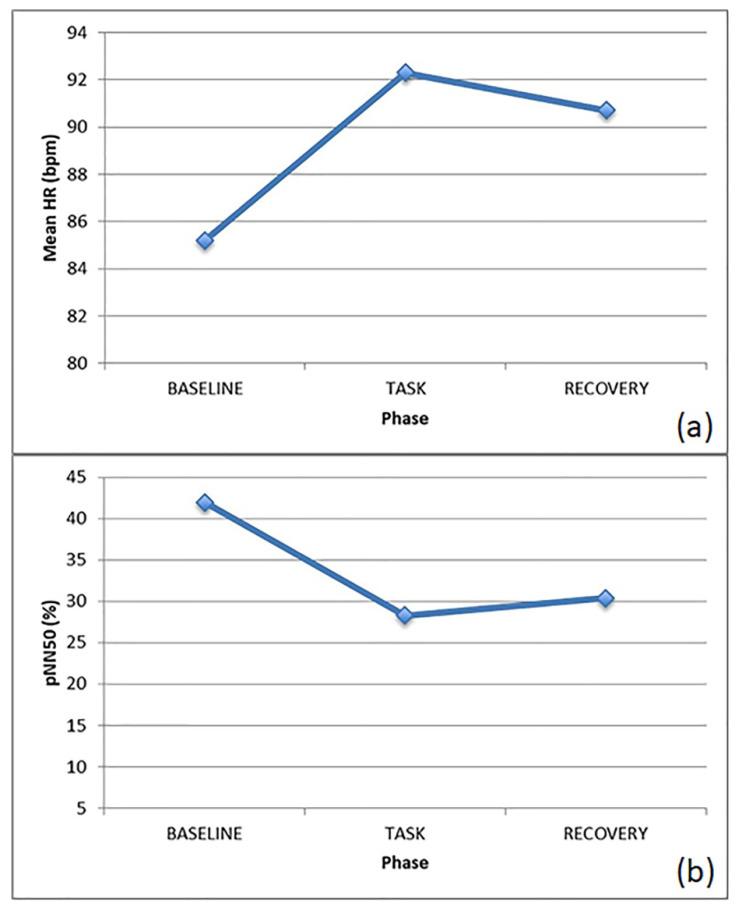
ECG features for J. during the various phases of the action on the represented: (**a**) HR, (**b**) pNN50.

**Figure 6 brainsci-10-00382-f006:**
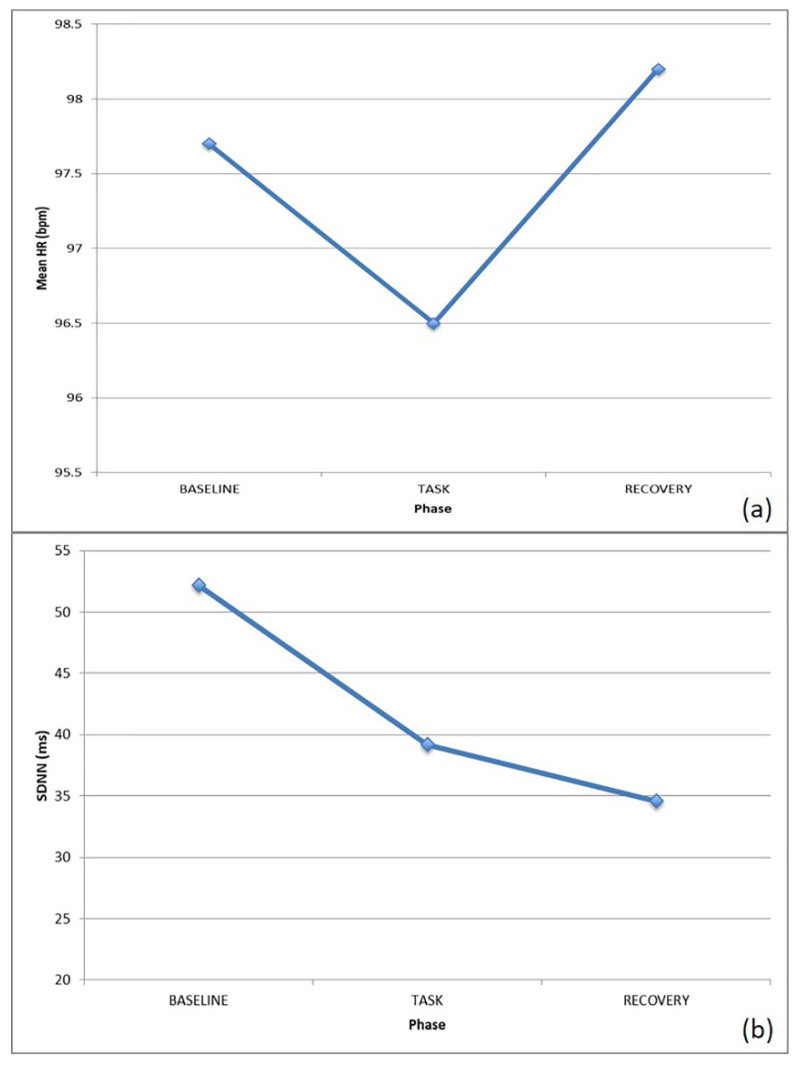
HR/HRV for B. during the various phases of the action on the represented: (**a**) HR, (**b**) SDNN.

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
