# Peer review of "Behavioral and Autonomic Responses in Treating Children with High-Functioning Autism Spectrum Disorder: Clinical and Phenomenological Insights from Two Case Reports"

_brainsci, 2020, doi:10.3390/brainsci10060382_

Round 1

Reviewer 1 Report

I am happy with the changes in the manuscript. Pleased to accept it for publication!

Reviewer 2 Report

I am happy to see that the authors have integrated all my comments and suggestions in the revised manuscript. I have no further revisions to suggest.

This manuscript is a resubmission of an earlier submission. The following is a list of the peer review reports and author responses from that submission.

Round 1

Reviewer 1 Report

The authors tested a new in-house approach to softener behavioral and physiological measures presented in Autistic children. The manuscript is well written and describes well the goal of the study. I would recommend adjustments in the methodology to allow others to replicate the study. My main comment is regarding to the conclusions driven by the study. In my opinion, no conclusion can be given about the use of c.m.i. as the authors can exclude that the results found were not just a matter of luck or maturation of the children that are in school age. I acknowledge that the authors suggest that there is no conclusion in this study but at the same time they say about the positive cognitive gains and benefits of c.m.i.. I believe that this study would work better as a small report than a full article unless the authors are able to provide a sample that is comparable with the 2 already in the study.

L100: please indicate the protocol number approved by the ethical committee. Also, provide an indication of where it was approved. Was in CNR? What does CNR mean?

I am wondering if this study would require a Clinical Trial number. Could you provide an explanation about this?

L103: please describe ADOS-G. Leiter-R and Progressive Color Matrices needs explanation.

L114: WPPSI-III and FSIQ need a proper description.

ADHD is not mention anywhere before L115.

Besides the information provided in the introduction, a proper description of c.m.i. is needed, perhaps in the methods. Please include all relevant information necessary for the reproduction of this experiment.

In addition, since c.m.i. is trademarked. Additional information regarding who owns it, where to find, how much it cost, and patent number is needed.

L127: please provide what it is included in the “exhaustive explanation” to the parents. You can add this in the suppl.

L133: since both kids are quite different, especially you think that one has ADHD, I would expect individual instructions to the parents. How these instructions were provided?

Were the kids given Spaghetti or Biscuit once a week for 10 months?

L133: how many times the therapists have contacted the parents after the initial meeting?

L304: it is undoubtedly the behavioral evolution of the kids tested. However, and here maybe my main concern, how can the author assure that these changes would not appear naturally over the course of 6-10 months at this age. I really a control, someone that undergo all the test without being trained to see the benefits of the c.m.i.

Funding, acknowledgments, and conflicts of interest were wrongly copied and pasted.

Reviewer 2 Report

The authors  have provided case study of two subjects diagnosed with high functioning autism . The study design demonstrated the importance of having personalized therapy to alleviated autism like symptoms in the subject. The findings are interesting.

However, I would like to suggest that the authors should include more subjects to increase the strength of this study.

Reviewer 3 Report

The study explores the communicative and interactive competence of two children diagnosed with High Functioning ASD, and also examines the nervous system responses of the two individuals after cognitive-motivational-individualized 6-month-training.This is an interesting study that will be of interest to academics in the field of neurodevelopmental disorders.

My main concerns relate to the following points:

Line 48: the authors should add references next to the point where they are talking about the presence of deficits in executive functions in ASD. Consider the following:

Pennington, B. F., Rogers, S. J., Bennetto, L., Griffith, E. M., Reed, D. T., & Shyu, V. (1997). Validity tests of the executive dysfunction hypothesis of autism. In J. Russell (Ed.), Autism as an executive disorder (pp. 143–178). Oxford: Oxford University Press.

Peristeri, E., Baldimtsi, E., Andreou, M., & Tsimpli, I. M. (2020). The Impact of Bilingualism on the Narrative Ability and the Executive Functions of Children with Autism Spectrum Disorders. Journal of Communication Disorders, 85, 105999.

 Lines 67-69: the authors should add references next to the point where they are talking about deficits in ASD children’s narrations. Consider the following:

King, D., Dockrell, J. E., and Stuart, M. (2013). Event narratives in 11-14 year olds with autistic spectrum disorder. Int. J. Lang. Commun. Disord. 48, 522-533.

Losh, M., and Capps, L. (2003). Narrative ability in high-functioning children with autism or Asperger’s syndrome. J. Autism Dev. Disord. 33, 239-251.

 Peristeri, E., Andreou, M., & Tsimpli, I. M. (2017).  Syntactic and Story Structure Complexity in the Narratives of High- and Low-Language Ability Children with Autism Spectrum Disorder. (Special Issue: Investigating Grammar in Autism Spectrum Disorders) Frontiers in Psychology. doi: 10.3389/fpsyg.2017.02027.  

Lines 50-53. The authors mention that the cognitive-motivational-individualized technique applies to subjects with verbal, cognitive and relational disability. How do they justify their choice to apply this technique to two individuals whose IQ lies within the normal range, i.e. they are not cognitively disable?

Discussion.  The overwhelming part of the discussion focuses on the study’s limitations, while the findings are not discussed at all. Discussion should be rewritten to interpret the findings in the Results section.